# Cannabis companies and the sponsorship of scientific research: A cross-sectional Canadian case study

**Quinn Grundy**[1]*, **Daphne Imahori**[2], **Shreya Mahajan**[2], **Gord Garner**[3], **Roberta Timothy**[4], **Abhimanyu Sud**[5,6,7], **Sophie Soklaridis**[2,5], **Daniel Z. Buchman**[2,4,8]

**1** Lawrence S. Bloomberg Faculty of Nursing, University of Toronto, Toronto, Canada, **2** Centre for Addiction and Mental Health (CAMH), Ontario, Canada, **3** Community Addictions Peer Support Association (CAPSA), Ottawa, Canada, **4** Dalla Lana School of Public Health, University of Toronto, Toronto, Canada, **5** Temerty Faculty of Medicine, University of Toronto, Toronto, Canada, **6** Institute of Health Policy, Management and Evaluation, University of Toronto, Toronto, Canada, **7** Humber River Hospital, Toronto, Canada, **8** University of Toronto Joint Centre for Bioethics, Toronto, Canada

* quinn.grundy@utoronto.ca

## Abstract

Corporations across sectors engage in the conduct, sponsorship, and dissemination of scientific research. Industry sponsorship of research, however, is associated with research agendas, outcomes, and conclusions that are favourable to the sponsor. The legalization of cannabis in Canada provides a useful case study to understand the nature and extent of the nascent cannabis industry's involvement in the production of scientific evidence as well as broader impacts on equity-oriented research agendas. We conducted a cross-sectional, descriptive, meta-research study to describe the characteristics of research that reports funding from, or author conflicts of interest with, Canadian cannabis companies. From May to August 2021, we sampled licensed, prominent Canadian cannabis companies, identified their subsidiaries, and searched each company name in the PubMed conflict of interest statement search interface. Authors of included articles disclosed research support from, or conflicts of interest with, Canadian cannabis companies. We included 156 articles: 82% included at least one author with a conflict of interest and 1/3 reported study support from a Canadian cannabis company. More than half of the sampled articles were not cannabis focused, however, a cannabis company was listed amongst other biomedical companies in the author disclosure statement. For articles with a cannabis focus, prevalent topics included cannabis as a treatment for a range of conditions (15/72, 21%), particularly chronic pain (6/72, 8%); as a tool in harm reduction related to other substance use (10/72, 14%); product safety (10/72, 14%); and preclinical animal studies (6/72, 8%). Demographics were underreported in empirical studies with human participants, but most included adults (76/84, 90%) and, where reported, predominantly white (32/39, 82%) and male (49/83, 59%) participants. The cannabis company-funded studies included people who used drugs (37%) and people prescribed medical cannabis (22%). Canadian cannabis companies may be analogous to peer industries such as pharmaceuticals, alcohol, tobacco, and food in the following three ways: sponsoring research related to product development, expanding indications of use, and supporting key opinion leaders. Given the recent legalization of cannabis in

**Data Availability Statement:** The full dataset, including sampled articles' metadata, are attached as S1 Dataset.

**Funding:** The study is funded by the Mental Health Commission of Canada through a Canadian Institutes of Health Research Catalyst Grant: Cannabis and Mental Health (Awarded to DZB and QG). The funder had no role in the study design, data collection, analysis, decision to publish, or preparation of the manuscript.

**Competing interests:** The authors have declared that no competing interests exist.

Canada, there is ample opportunity to create a policy climate that can mitigate the harms of criminalization as well as impacts of the "funding effect" on research integrity, research agendas, and the evidence base available for decision-making, while promoting high-priority and equity-oriented independent research.

## Introduction

In October 2018, Canada became the second country and first G7 nation to legalize the cultivation, possession, acquisition, and consumption of cannabis. In November 2018, two Canadian universities announced cannabis company-sponsored research chairs for topics including the medicinal potential of cannabinoids, the genetic properties of cannabis plants related to cold climate cultivation, laboratory-based methods for producing cannabis compounds, and the role of cannabis in addressing the opioid overdose crisis [1, 2].

Corporations across sectors engage in the conduct, sponsorship, and dissemination of scientific research. A "funding effect" is well-established across sectors and scientific fields of research, wherein industry sponsorship is statistically and positively associated with research outcomes and conclusions that are favourable to the sponsor, independent of methodological quality [3–5]. The result can be a distortion of the evidence base needed to effectively evaluate the benefits and harms of the industry's products, processes, or activities [6–8]. The mechanisms for the funding effect are complex and likely include publication bias (wherein sponsors suppress studies with unfavourable results) and methodological choices around research or experimental design that produce more favourable outcomes [5].

There is also evidence that industry sponsorship is associated with shifts in research agendas, which entails defining the research purpose, prioritizing lines of inquiry, and framing research questions [9]. Life scientists across multiple surveys reported the perception that industry funding is associated with shifts in research agendas toward more applied research with commercial application [9]. Scientists report ambivalence toward commercialization of research, but also frequently accept the inevitability of industry partnerships within the context of funding pressures, precarious academic labour markets, and the culture of biomedical research [10]. Scientists' perceptions are confirmed by a body of meta-research associating industry sponsorship with patterns in research agendas that focus on products, processes, or activities that can be commercialized and marketed, and topics that support the industry's policy and legal positions [9]. For example, an analysis of randomized controlled trials included in systematic reviews of nutrition interventions to address obesity found that industry-sponsored trials were more likely than non-industry-sponsored to trials to focus on the manipulation of specific nutrients instead of interventions targeting dietary behaviours or patterns [11]. The available evidence may then be disproportionately focused on policy solutions that favour commercial interests (such as food fortification) rather than those that do not (such as taxes, regulation of advertising, or addressing food processing) [11].

Corporations also seek to shape evidence and policy environments through financial relationships with key opinion leaders—respected, credible, and influential experts within a field —that hold perspectives that support company positions [6, 12]. By providing key opinion leaders with paid consultancies, advisory board memberships, and speaking engagements, corporate sponsors are able to amplify the corporation's positions. For example, research articles where authors disclose such conflicts of interest with company sponsors are more likely to be cited or receive media attention [13].

While corporate research and development has undoubtedly contributed to innovation across sectors, concerns remain regarding the impact of industry sponsorship on research integrity and the evidence base used to inform regulatory and policy decisions. Emerging evidence suggests that the nascent cannabis industry is engaging in partnerships with academic institutions and researchers and publicizing research-related claims in their cannabis marketing [14]. However, an analysis of research-related claims in the marketing materials of six major medical cannabis companies found that statements implied the safety and efficacy of their products for a range of serious health conditions, but none were based on published research that could provide causal evidence (e.g. a randomized controlled trial) [14].

The legalization of cannabis in Canada provides a useful case study of cannabis industry involvement in the production of scientific and health-related evidence [6]. Due to the way that cannabis is regulated in Canada (i.e. as a drug), researchers are reliant upon cannabis companies to produce and provide cannabis for research purposes following Good Manufacturing Purposes, and are prohibited from conducting clinical trials using cannabis products purchased through commercial retail outlets as they do not meet these standards [15]. Thus, Canada's regulations require academic-industry partnerships for certain types of cannabis-related research and preclude independent evaluation of these products [15].

The International Committee of Medical Journal Editors recommends that all articles be published with statements detailing sources of study support and authors' relationships with for-profit companies whose interests may be affected by the content [16]. Since 2017, these conflict of interest statements, as provided by publishers, are searchable within PubMed records [17]. Thus, the aim of this cross-sectional meta-research study was to: 1) identify published research where authors disclosed funding or financial relationships with Canadian cannabis companies; 2) describe the nature of cannabis company sponsorship; 3) describe characteristics of research with cannabis company sponsorship including research type and focus.

The Canadian context not only offers an important case study to understand patterns in sponsorship of research by an emerging industry, but also to question the broader impacts of industry sponsorship of research on social and health equity. There has been no research, to our knowledge, exploring the relationships between industry sponsorship of research and characteristics of research agendas or research design that might address the priorities of groups disproportionately affected by the industry's activities or industry regulation. Decades-long prohibition and systemic racism in Canada contributes to the disproportionate criminalization of Black and Indigenous people for the possession, production, and sale of cannabis [18]. The Canadian government, at the time of this writing, has yet to address the issue of cannabis amnesty or how to ensure equitable access to the economic benefits of the emerging industry [18–20]. The legacy of criminalization in Canada has also generated critical research gaps that perpetuate stigma and misinformation, ultimately undermining quality of care and support [21]. In response, leaders within Black communities in Canada have proposed a cannabis research agenda that is Black-led, addresses structural issues, and collects and reports race-based data [21]. Thus, for research with human participants, a final study aim was to characterize the degree of demographic reporting (including race-based data collection) and demographics of included participants.

## Methods

We conducted a cross-sectional, meta-research case study of the research activities of Canadian cannabis companies. We report the methods according to the STROBE guideline (**S1 Checklist**).

## Sampling and data sources

We employed criterion purposeful sampling to identify cannabis companies that grow, process, market, and/or distribute cannabis products and were licensed within Canada, operating either directly or through a subsidiary. In May 2021, we identified a preliminary list of prominent Canadian cannabis companies indexed by New Cannabis Ventures, an industry investor news platform, to serve as the sampling frame. We then grouped listed companies into 'families' based on shared ownership and verified parent companies' licenses through the Health Canada database. We compiled a comprehensive list of all subsidiary companies within a family using financial statements, Crunchbase (a public database of companies), and press releases.

On August 20, 2021 we used the PubMed advanced search function to search the conflict of interest field for the name of each parent and subsidiary company (**S1 and S2 Tables**). The conflict of interest search interface in PubMed does not allow for Boolean logic or multi-word searches. To increase the relevance of search results, we searched company's base names (e.g. Canopy Growth and not Canopy Growth Corp.) and excluded subsidiaries that had only numeric identification (e.g., 195729 Ontario Inc.). We screened all results manually, excluding entries that did not pertain to cannabis companies or were ambiguous (e.g. we included information pertaining to "Peloton Therapeutics" but excluded articles that only reference "Peloton," which could refer to Peloton Therapeutics or the exercise equipment company). We included articles where at least one author declared a conflict of interest or study support from a Canadian cannabis company that grows, processes, markets, and/or distributes cannabis products within Canada. There were no restrictions by date of publication, article type, or language.

## Data extraction and analysis

Using the R version 4.1.0 [22] and the package easyPubMed [version 2.13] [23], we automatically extracted article metadata, as indexed by PubMed, which included: author names, author affiliations, title, abstract, journal, article type, DOI, and verbatim conflict of interest statement. We also used the packages rlist [24], tidyr [25], stringr [26], and dplyr [27] for data cleaning.

Two coders independently, manually extracted data from articles' full texts using a coding manual adapted from a previous project [13] and based on the ICMJE definition and typology of conflict of interest (**S3 Table**) [16]. A third author resolved any coding discrepancies. We relied on PubMed English abstracts and titles and Google translate to extract data from articles written in languages other than English.

We coded articles by article type into one of three categories: 1) empirical research (any empirical research such as randomized controlled trials, observational studies, case reports and qualitative research); 2) commentaries, editorials, or narrative reviews (articles that did not report original data and did not report a methodology); and 3) systematic reviews and meta-analyses (identified through reported methodologies).

We screened conflict of interest and funding declaration statements and identified all named cannabis companies including those in the original search strategy. We then additionally identified any other Canadian or international companies that grow, process, market, or distribute non-medical or medical cannabis, or provide services to the cannabis industry (e.g. cannabis marketing, cannabis news). We treated study funding and author conflict of interest separately. We coded the presence and type of study funding and conflict of interest following the ICMJE typology, for cannabis and non-cannabis companies, following authors' verbatim declarations. For example, we coded a conflict of interest as "present" where one or more of

the authors disclosed a conflict of interest of any type; as "absent" if all authors made an explicit statement of no conflict of interest; and as "missing" where there was no statement about the presence or absence of author conflict of interest.

We assessed whether the article was cannabis-focused (assessed as mentioning cannabis in the title or abstract). Two authors then inductively and iteratively identified and thematically categorized the main topic of the article, assessed as the focus of the study reported in the title and abstract, treating these categories as mutually exclusive.

We categorized empirical articles as plant or animal research using a dichotomous yes/no variable. For empirical articles with human participants, we determined whether the study was interventional (defined as prospective intervention studies including any trials or experiments with cannabis), whether the article reported demographic categories (including age, sex or gender, race or ethnicity, sexual orientation, socioeconomic status, and education level), and the demographics of included participants, where reported.

We piloted the coding manual on a random 10% sample of included articles, with 85% agreement. We elected to code the remaining articles in pairs of authors working independently to ensure comprehensiveness and accuracy, with discrepancies reviewed by a third author and final coding reached by consensus. We analyzed results using descriptive statistics in Excel, using the article as the unit of analysis.

## Results

To characterize the research activities of the emerging Canadian cannabis industry, in May 2021, we began by extracting a list of 50 prominent Canadian cannabis companies indexed by New Cannabis Ventures, an industry investor news platform, to serve as our sampling frame [28]. We identified all the parent and subsidiary companies operating and licensed in Canada associated with these 50 companies and conducted 254 unique searches for each parent and subsidiary company name using the PubMed advanced search builder. **Fig 1** outlines the sampling process.

Following screening of the search results, we included 156 articles in which at least one author declared cannabis company research support (financial or non-financial) or disclosed a relationship with one of the Canadian cannabis companies that grow, manufacture, market, or distribute cannabis products included in our original search. Most articles (150/156, 96%) were published since legalization (i.e. 2018) and reported empirical research (110/156, 71%) (**Table 1**). The full dataset is available in **S1 Dataset**.

Authors of sampled articles reported relationships with only 6% (15/254) of the Canadian cannabis companies (parent and subsidiary) included in the search. However, most of the disclosure statements (95/156, 61%) reported author ties to multiple cannabis companies, ranging from 1–20 associations per article. In addition to disclosures of research support or author relationships with the Canadian cannabis companies included in our original search (n = 15), we identified an additional 68 cannabis-related companies within the disclosure statements of the included articles. While 18% (12/68) of these additional companies were Canadian growers, processors, or retailers, the majority were based internationally; 60% (41/68) were not Canadian, 35% (24/68) were pharmaceutical companies or distributors of prescription cannabis products, and 18% (12/68) provided services to cannabis companies such as software, marketing, or brand development. Despite the number and range of cannabis industry entities named in author disclosure statements, five corporations dominated with each named in more than 10% of the disclosure statements: Tilray Brands Inc. (80/156, 51%), Canopy Growth Corporation (62/156, 40%), and Aurora Cannabis Inc. (17/156, 11%), which were included in the search strategy, plus GW Pharmaceuticals (20/156, 13%) and Zynerba Pharmaceuticals, Inc.

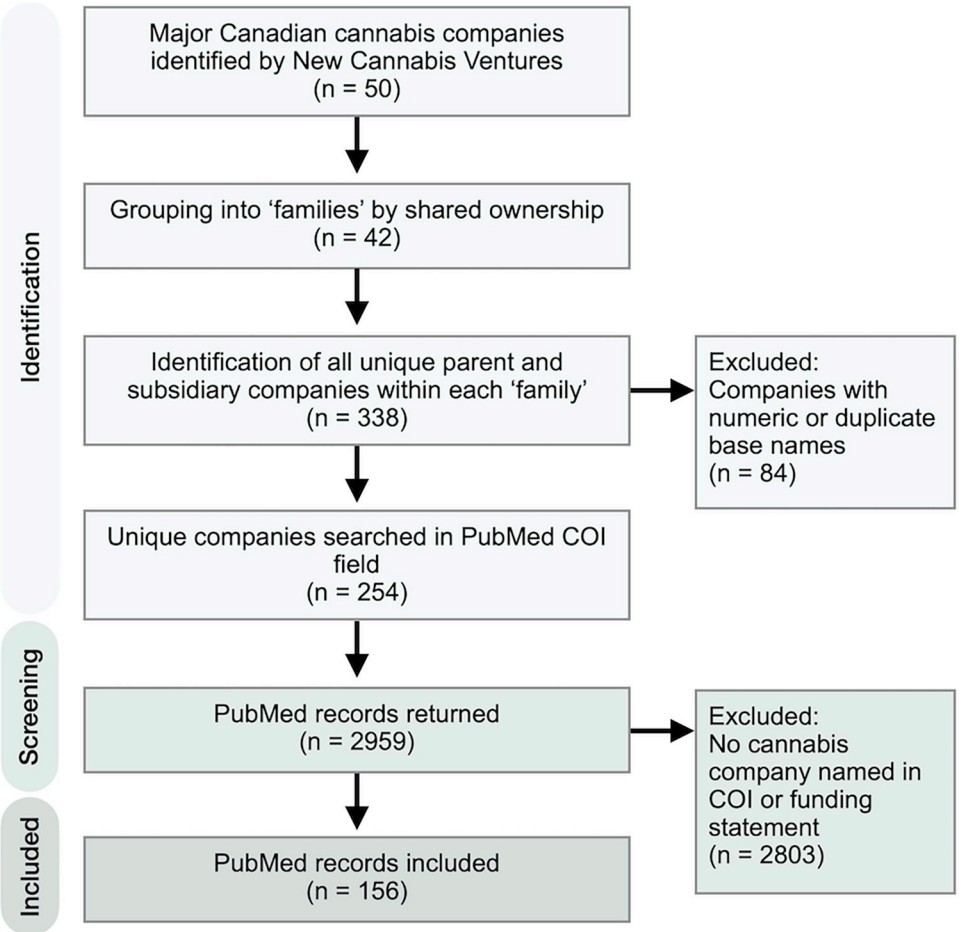

**Fig 1. Study identification and sampling flow diagram.**

(18/156, 18%), which were identified when we analyzed included articles' disclosure statements.

In 82% (128/156) of article disclosure statements, at least one author disclosed a conflict of interest with at least one cannabis company, defined per the ICMJE as any relationship between an individual author and a for-profit or not-for-profit third parties whose interests may be affected by the content of the article (see Table 1) [16]. Most often, authors reported receipt of personal fees from cannabis companies for activities such as consulting or advisory board membership (95/128, 74%). In 65% (101/156) of the disclosure statements, author teams declared conflicts of interest with both cannabis and non-cannabis companies.

In contrast to the prevalence of disclosed author conflicts of interest, only 35% of articles (54/156) reported study funding from a cannabis company, defined as any financial or non-financial support for the work reported in the article. Prevalent forms of cannabis company funding included cannabis company employees as co-authors (33/54, 61%) and research grants (25/54, 46%). Nearly ¼ of the articles reported nonfinancial support (13/55, 24%), such as the provision of product for evaluation or access to data. Twelve percent (18/156) of articles reported exclusive funding by cannabis companies.

Over half of the included articles (84/156, 54%) did not have a cannabis focus, meaning cannabis was not mentioned in the title nor abstract. For example, many articles reporting

**Table 1. Article characteristics overall and by type of cannabis-company relationship.**

| Cannabis company relationship | Any (N = 156) N (%)[a] | Author COI N (%)[b] | Study support N (%)[b] | Both N (%)[b] |
|---|---|---|---|---|
| **Article type** | | | | |
| Empirical[c] | 110 (71%) | 84 (76%) | 44 (40%) | 19 (17%) |
| Commentary[d] | 36 (23%) | 34 (94%) | 8 (22%) | 7 (19%) |
| Systematic review[e] | 10 (6%) | 10 (100%) | 3 (30%) | 3 (30%) |
| **Author location** | | | | |
| Canada and USA | 90 (58%) | 67 (74%) | 44 (49%) | 21 (23%) |
| Global | 36 (23%) | 32 (89%) | 4 (11%) | 2 (6%) |
| Europe | 24 (15%) | 23 (96%) | 4 (17%) | 3 (13%) |
| Australia and Asia-Pacific | 6 (4%) | 6 (100%) | 2 (33%) | 2 (33%) |
| **Study subjects** | | | | |
| Humans | 138 (88%) | 118 (86%) | 41 (30%) | 23 (17%) |
| Animals | 11 (7%) | 7 (64%) | 7 (64%) | 3 (27%) |
| Plants | 7 (4%) | 3 (43%) | 6 (86%) | 2 (29%) |
| **Cannabis-focused content[f]** | | | | |
| No | 84 (54%) | 72 (86%) | 12 (14%) | 2 (2%) |
| Yes | 72 (46%) | 56 (78%) | 42 (58%) | 26 (36%) |

[a]Percentages are column percentages out of total sample of n = 156

[b]Percentages are row percentages

[c]Reports of original research, published protocols, or case reports

[d]Commentaries, narrative reviews, editorials, opinion pieces

[e]Systematic reviews and meta-analyses

[f]Assessed as mention of cannabis in title or abstract

empirical research were authored by multi-author teams who provided lengthy disclosure statements reporting conflicts of interest with many pharmaceutical and biomedical companies, one of which was a cannabis company. Despite these occurrences, a higher proportion of articles with a cannabis focus (42/72, 58%) received cannabis company funding than those without (12/84, 14%).

Among the 72 articles that did have a cannabis focus, prevalent topics included: cannabis as a treatment for a range of conditions (15/72, 21%), particularly chronic pain (6/72, 8%); cannabis as a tool in harm reduction related to opioid or other substance use (10/72, 14%); the safety and risk of harm of cannabis-related products (e.g., effect on driving, tolerability and drug interactions) (10/72, 14%); and preclinical animal studies using cannabis-related products (**Table 2**).

Among articles with a cannabis focus, 25 (35%) included cannabis company employee authors; 17 (24%) received study support from a cannabis company in the form of a research grant; and 12 (17%) received non-financial support in the form of product, devices, data, technical, or writing support. Authors of the majority of cannabis-focused articles (56/72, 78%) declared a conflict of interest with a cannabis company; prevalent declarations included receipt of personal fees from cannabis companies (47/72, 65%), cannabis company research sponsorship for past projects (16/72, 22%), and ownership interest or stocks in cannabis companies, including the sponsor (15/72, 21%).

The sample of 156 included 87 articles detailing empirical studies with human participants, the majority of which were conducted in Canada or the United States (57/87, 66%) (**Table 3**). Among empirical studies (n = 87), 28 (32%) prospectively introduced an intervention, and 12 (14%) included a cannabis-focused intervention. Most of the cannabis-focused interventional

**Table 2. Content focus of cannabis-focused articles (n = 72).**

| Article topic | N | % |
|---|---|---|
| Cannabis product as a treatment | 16 | 22 |
| Chronic pain | 6 | 8 |
| Chronic tic disorders | 2 | 3 |
| COVID-19 | 1 | 1 |
| Insomnia | 1 | 1 |
| Memory/processing disorders | 1 | 1 |
| Patient-reported outcomes | 1 | 1 |
| Progression of Amyotrophic lateral sclerosis | 1 | 1 |
| PTSD | 1 | 1 |
| Seizure disorders | 1 | 1 |
| Chemo induced nausea and vomiting | 1 | 1 |
| Harm reduction | 10 | 14 |
| Substitute for other licit and illicit drugs | 6 | 8 |
| Opioid use | 3 | 4 |
| Policy | 1 | 1 |
| Safety/harms | 10 | 14 |
| Impaired driving | 3 | 4 |
| Safety, tolerability, and drug interactions | 3 | 4 |
| Neurocognitive adverse effects | 2 | 3 |
| Impact on healthcare utilization | 1 | 1 |
| Product labeling and composition | 1 | 1 |
| Preclinical (animal model) | 6 | 8 |
| Patterns in cannabis use and preferences | 5 | 7 |
| Agriculture/horticulture | 4 | 6 |
| Health Care Professional education about cannabis (dispensary staff) | 4 | 6 |
| Pharmacokinetics/physiology | 4 | 6 |
| Relationship between cannabis use and mental health conditions | 4 | 6 |
| Legalization effects | 3 | 4 |
| Impact of opioid agonist therapy on cannabis and other drug use | 2 | 3 |
| Treatment of cannabis use disorder | 2 | 3 |
| Relationship between cannabis use and tobacco use | 1 | 1 |
| Product testing methods | 1 | 1 |

studies (7/12, 58%) received cannabis company research support, mostly comprised of donated products.

Most empirical studies with human participants reported participants' age (84/87, 97%) and sex and/or gender (83/87, 95%). A minority of studies, ranging from 2% to 45%, reported at least one of the following: race and/or ethnicity, sexual orientation, socioeconomic status, or education level. Of those reporting demographic details, most empirical studies with human participants included adults (76/84, 90%), and samples with more than 50% of participants identifying as white (32/39, 82%), male (49/83, 59%), and heterosexual (2/2, 100%). Cannabis-focused and cannabis company-funded empirical studies included people who use drugs (13% and 37%, respectively), and people prescribed medical cannabis (29% and 22%, respectively).

## Discussion

This descriptive, cross-sectional study provides evidence of Canadian cannabis company involvement in the conduct and sponsorship of research and engagement with key opinion

**Table 3. Demographic reporting and inclusion in empirical studies with human participants.**

| Category | Empirical studies (N = 87) n (%) |
|---|---|
| **Demographic reporting** | |
| Reported sampled age | 84 (97%) |
| Adults (> 18 years) | 76 (90%) |
| Children only | 5 (6%) |
| Young adults (18–24 years) only | 2 (2%) |
| Older adults (>65 years) | 1 (1%) |
| Reported sex and/or gender | 83 (95%) |
| Reported racialization and/or ethnicity | 39 (45%) |
| Reported socioeconomic status | 30 (34%) |
| Reported education level | 22 (25%) |
| Reported sexual orientation | 2 (2%) |
| **Description of included participants** | |
| People prescribed medical cannabis | 11 (13%) |
| People who use drugs | 11 (13%) |
| People who use cannabis non-medically | 4 (5%) |
| People prescribed opioid agonist therapy | 4 (5%) |
| People living with: | |
| Cancer | 8 (9%) |
| Epilepsy, Dravet Syndrome, or CDKL5 deficiency disorder | 8 (9%) |
| Acute and chronic pain | 8 (9%) |
| Neuromuscular and movement disorders | 6 (7%) |
| HIV/AIDS | 3 (3%) |
| COPD | 3 (3%) |
| Inflammatory and autoimmune disease | 2 (2%) |
| Acute infection | 1 (1%) |
| Varicose veins | 1 (1%) |
| Sleepwalking | 1 (1%) |
| Gambling addiction | 1 (1%) |
| Chronic disease risk factors | 1 (1%) |
| Cannabis use disorder | 1 (1%) |
| Healthy volunteers | 7 (8%) |
| Bereaved family members or sudden death in children | 3 (3%) |
| Military veterans | 2 (2%) |
| Cannabis dispensary staff | 1 (1%) |

leaders around the time of legalization of cannabis in Canada. Authors of sampled articles disclosed relationships with a wide variety of cannabis companies, but two companies—Tilray and Canopy Growth—had a dominant presence, which likely reflects their dominance within the market in terms of revenue and market capitalization [29, 30]. Most of the sampled articles reported empirical research, however, only 10% of sampled articles reported the results of interventional studies, which prospectively evaluated a cannabis product intervention. This finding is consistent with recent research examining the marketing materials of medical cannabis companies citing research data, which found that the cannabis companies rarely referenced studies with designs that could provide high-quality evidence of product safety or effectiveness [11]. This likely also reflects the highly regulated environment in Canada related to the conduct of cannabis clinical trials, which makes researchers reliant on the cannabis industry to produce and supply research-grade cannabis products [15].

Among sampled studies with an explicit focus on cannabis, the largest proportion focused on cannabis as a treatment for a range of chronic conditions, including chronic pain, and as a harm reduction strategy for people who use drugs. Cannabis is advocated as a potential substitute for opioids in the treatment of pain, an opioid reduction strategy, and an adjunct to opioid agonist therapy [31]. The tobacco, pharmaceutical, and alcohol industries have also marketed products as less harmful, or even an antidote, to the products with addictive potential that define their industries [6]. These findings suggest that the cannabis industry has entered discussions around harm reduction through research-related partnerships. Further independent research, led-by or conducted in collaboration with people who use substances and who are living with mental health conditions, is particularly needed to inform policy discussions about cannabis within a harm reduction context.

Consistent with issues related to representation and diversity in clinical research population samples, sampled empirical studies with human participants predominantly included white men, and further, demographic details were altogether underreported. Meanwhile, community leaders have underscored the need for cannabis-related research that is undertaken in collaboration with, or led by Black communities and that use methodologies which incorporate an intersectional lens, for example [21]. Given the disproportionate harms experienced by Black and Indigenous communities from the activities of corporations and the promotion of products harmful to human health (e.g., [32]), scrutiny is required to examine the role of industry sponsorship in equity-oriented research and further support is needed for independent research.

## Strengths and limitations

In searching conflict of interest statements directly, we undertook an innovative use of the PubMed search interface. However, the search was thus restricted to biomedical literature and was not particularly sensitive nor specific, necessitating manual screening and resulting in identification of many companies not included in the original search. In restricting our search to Canadian cannabis companies, we have likely captured a fraction of the research literature that has ties to cannabis companies. Further, it is unknown the extent to which PubMed metadata related to conflict of interest is complete or comprehensive in terms of indexed journals; thus, our study likely also underestimates the extent of Canadian cannabis company research sponsorship or financial relationships with researchers. For example, previous research analyzing the conflict of interest statements of articles published between 2016 and 2018 and indexed in PubMed found that only one of the highest h-index medical journals reported conflict of interest information to PubMed [33]. Thus, this descriptive analysis necessarily provides a partial perspective of the phenomenon of Canadian cannabis company research activities. We did not assess the sampled articles' specific study design, direction of results, or concordance with conclusions; thus, we are unable to assess sponsorship biases in this sample.

## Conclusions

Our findings suggest that Canadian cannabis companies, around the time of legalization of their product in Canada, conducted research activities akin to peer industries such as pharmaceuticals, tobacco, alcohol, and food in three ways: sponsoring research related to product development and testing, expanding indications of use, and financially supporting key opinion leaders. Given the recent legalization of cannabis in Canada, there is ample opportunity to create a policy climate that can mitigate the harmful effects of criminalization as well as the "funding effect" on research integrity, research agendas, and the evidence base available for decision-making. As an emerging industry, the Canadian cannabis industry has experienced a

high degree of flux and the *Cannabis Act* is currently under review. In a recent article published in a national Canadian newspaper, researchers lamented the scarcity of pharmaceutical-grade cannabis for research purposes—the head of the industry association reported that growers had largely stopped funding research in light of unattractive profit margins and the regulatory climate [34]. Thus, policymakers need to consider how to ensure that necessary research happens at all, while also prioritizing independence and community engagement in the research process. As more jurisdictions globally consider cannabis legalization, policymakers in these contexts should build in levers to ensure high quality, independent cannabis research.

## Supporting information

**S1 Checklist. STROBE reporting checklist for cross sectional studies.**
(DOCX)

**S1 Table. Sampling and search strategies.**
(DOCX)

**S2 Table. List of PubMed searches.**
(DOCX)

**S3 Table. Coding manual.**
(DOCX)

**S1 Dataset.**
(XLSX)

## Author Contributions

**Conceptualization:** Quinn Grundy, Gord Garner, Roberta Timothy, Abhimanyu Sud, Sophie Soklaridis, Daniel Z. Buchman.

**Data curation:** Quinn Grundy, Daphne Imahori, Shreya Mahajan.

**Formal analysis:** Quinn Grundy, Daphne Imahori, Gord Garner, Roberta Timothy, Abhimanyu Sud, Sophie Soklaridis, Daniel Z. Buchman.

**Funding acquisition:** Quinn Grundy, Daniel Z. Buchman.

**Investigation:** Quinn Grundy, Daphne Imahori, Shreya Mahajan.

**Methodology:** Quinn Grundy, Roberta Timothy, Daniel Z. Buchman.

**Project administration:** Shreya Mahajan.

**Supervision:** Quinn Grundy, Roberta Timothy, Daniel Z. Buchman.

**Validation:** Gord Garner.

**Writing – original draft:** Quinn Grundy.

**Writing – review & editing:** Daphne Imahori, Shreya Mahajan, Gord Garner, Roberta Timothy, Abhimanyu Sud, Sophie Soklaridis, Daniel Z. Buchman.

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
