## [Decision Letter · Decision Letter 0]

6 Dec 2022

PONE-D-22-29784Canadian cannabis companies and the sponsorship of scientific research: A cross-sectional Canadian case studyPLOS ONE

Dear Dr. Grundy,

Thank you for submitting your manuscript to PLOS ONE. After careful consideration, we feel that it has merit but does not fully meet PLOS ONE’s publication criteria as it currently stands. Therefore, we invite you to submit a revised version of the manuscript that addresses the points raised during the review process.

I recommend that it should be revised taking into account the changes requested by the reviewers. Since the requested changes include valuable and constructive reviews, I would like to give you a chance to revise your manuscript. The revised manuscript will undergo the next round of review by same reviewers.

We look forward to receiving your revised manuscript.

Kind regards,

Baogui Xin, Ph.D.

Academic Editor

PLOS ONE

Journal Requirements:

"The authors report no conflicts of interest."

Reviewers' comments:

Reviewer's Responses to Questions

**Comments to the Author**

1. Is the manuscript technically sound, and do the data support the conclusions?

Reviewer #1: Yes

Reviewer #2: Yes

2. Has the statistical analysis been performed appropriately and rigorously? 

Reviewer #1: Yes

Reviewer #2: Yes

3. Have the authors made all data underlying the findings in their manuscript fully available?

Reviewer #1: No

Reviewer #2: No

4. Is the manuscript presented in an intelligible fashion and written in standard English?

Reviewer #1: Yes

Reviewer #2: Yes

5. Review Comments to the Author

Reviewer #1: The methods are rigorous and best-practice for meta-research. The paper is well-written and results are not overstated.

My only recommendations is that I think the data availability statement is inadequate for PLOS One. Data should be deposited in a public repository. Given this is meta-research, I don't see any barrier to this (i.e. no patient information, IRB oversight, etc.)

Reviewer #2: “Canadian cannabis companies and the sponsorship of scientific 1 research: A cross-sectional Canadian case study” offers a systematic study of the study sponsorship and conflicts of interest supported by the Canadian cannabis sector. The study deploys a novel search protocol focusing on identified Canadian cannabis firms and PubMed-indexed conflicts of interest statements. The study collected 156 articles with a named cannabis company in the COI statement conducted three data collection exercises. The first collected key data on all articles and focused on study type (research, commentary, review), author location, COI disclosures, study subjects (human, plants, non-human animals) and the presence or absence of specific cannabis content in the study. A subsequent content analysis of the 72 cannabis-focused articles identifies article topics including conditions addressed (where appropriate) and other article foci (e.g., harm reduction, cannabis harms, preclinical studies, etc). The final analysis extracts participant demographics data for the 87 collected human studies. This is one of the first systematic studies of COI in research on medical applications of cannabis. The study design is appropriate and rigorous. The submitted results provide an important foundation for future research in this area. I recommend publication following a few revisions:

•The content analysis would be improved if combined with the COI analysis. Specifically, given the overall focus of the article, I want to know more about which article topics were associated with cannabis sponsorship or COIs. This information could be added to Table 2 and discussed in the results section.

•As replicability with R can be a special challenge, I would recommend describing the R version and key packages (and versions) in a bit more detail.

•The limitations should be more specific about the gaps in coverage for COI reporting on PubMed where open access journals are over-represented. Highlighting this limitation is important to showcase the likelihood that there are even greater cannabis COI rates than currently observable in the current study.

•Inter-rater reliability scores using a chance-correcting statistics, reported alongside percent agreement, would provide readers with a more comprehensive understanding of the agreement picture.

•All data should be made available in the supplementary materials or in a data repository per Plos One guidelines.

6. PLOS authors have the option to publish the peer review history of their article (what does this mean?). If published, this will include your full peer review and any attached files.

Reviewer #1: No

Reviewer #2: No

---

## [Author Response · Author response to Decision Letter 0]

14 Dec 2022

Thank you for the encouraging and constructive reviews. See our attached Response to Reviewers, and note that we've added our dataset in Supplementary File 5 as requested.

---

## [Editor Report · Decision Letter 1]

20 Dec 2022

Cannabis companies and the sponsorship of scientific research: A cross-sectional Canadian case study

PONE-D-22-29784R1

Dear Dr. Grundy,

We’re pleased to inform you that your manuscript has been judged scientifically suitable for publication and will be formally accepted for publication once it meets all outstanding technical requirements.

Kind regards,

Baogui Xin, Ph.D.

Academic Editor

PLOS ONE
---

## [Editor Report · Acceptance letter]

26 Dec 2022

PONE-D-22-29784R1 

Cannabis companies and the sponsorship of scientific research: A cross-sectional Canadian case study 

Dear Dr. Grundy:

I'm pleased to inform you that your manuscript has been deemed suitable for publication in PLOS ONE. Congratulations! Your manuscript is now with our production department. 

Kind regards, 

on behalf of

Professor Baogui Xin 

Academic Editor

PLOS ONE